# Good Trees: Pruning Random Forests Without Compromise

## Abstract

Random forests are a powerful ensemble-based machine learning tool that combines multiple decision trees to form a better predictor than that based on the individual trees. Because each random forest prediction utilizes every tree's prediction, computation scales linearly as the ensemble size increases. Motivated by the idea that an intelligently selected subset of trees can perform comparably to the random forest it is derived from, we propose two novel algorithms to reduce the number of trees in a forest without compromising performance: tree slices and pruned groups. Tree slices select all trees within a specific performance range, while pruned groups use a representative sample of the forest to make a weighed approximation of the forest's performance. While performance can vary between datasets, the results are promising and suggest certain workflows can be greatly improved by these techniques.

## 1 Introduction

Decision trees take a divide-and-conquer approach to classification and regression, where data is repeatedly divided based on its features. Random forest is an ensemble machine learning model that aggregates the predictions of multiple decision trees to form a single prediction. Because each tree uses a random subset of features at each node split, different trees within a forest can have significant performance differences. The motivating idea behind random forests is that errors in one tree are offset by the decisions of other trees; aggregating individual predictors reduces variance, potentially improving overall prediction (Breiman, 1998; Segal & Xiao, 2011; Genuer & Poggi, 2020; Sagi & Rokach, 2018; Curth et al., 2024).

A fundamental limitation with random forests is their ensemble nature; once the model is fit and a forest generated, each subsequent prediction requires computing the predictions of every tree in the forest. Computation time and model storage scales linearly with ensemble size, which makes the standard random forest model computationally expensive and time consuming for inference (Donges et al., 2024; Nan et al., 2016; Louppe, 2015). In high-volume production environments like fraud detection and product recommendation, this can be a serious problem (Afriyie et al., 2023; Donges et al., 2024; Everingham et al., 2016). Furthermore, Grinsztajn et al. (2022) finds that on tabular data, neural networks tend to smooth too much, relative to tree-based methods. This raises the possibility that ensemble pruning may allow random forests to outperform neural networks in both computational speed and accuracy. Thus, developing effective methods for pruning a forest (ie, reducing the number of trees in a forest) is highly valuable.

The literature differentiates between *static* pruning (a fixed forest is first generated, then some of its trees are removed) and a *dynamic* approach (in which pruning is performed in concert with the forest generation). Static pruning follows the "overproduce and choose" paradigm, where the forest overproduces a number of trees, then selects a subset of trees to use for prediction (Kulkarni & Sinha, 2012). An example of dynamic pruning can be seen in Tripoliti et al. (2010), which uses online curve fitting to continuously add trees until some criterion – such as fitted value or accuracy – is met; for example, if a forest needs an overall accuracy of $X\%$, then new trees would be created – and added to the forest – until that accuracy goal is achieved.

This paper focuses exclusively on static pruning, or the selection and usage of a subset of trees from a pre-existing forest. Note that the focus is only on *forest* pruning (reducing the number of trees in the ensemble) and not decision tree pruning (reducing the number of nodes in a tree) or feature pruning (selecting and

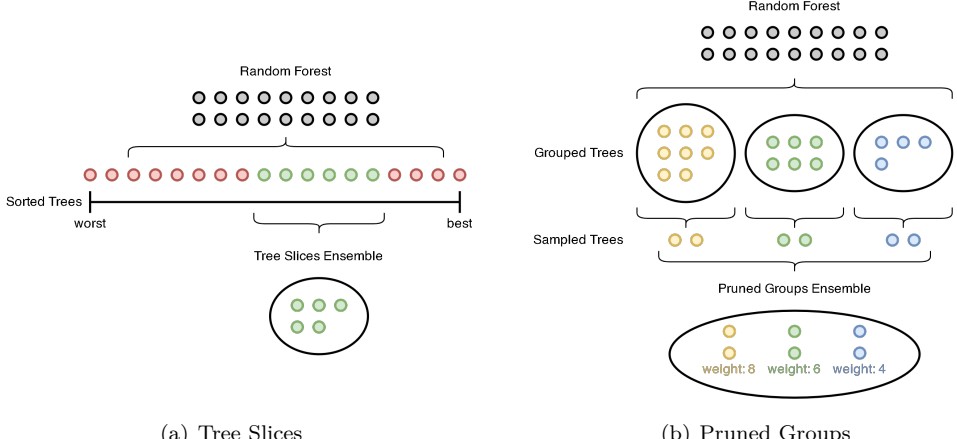

Figure 1: Overview of Tree Slices and Pruned Groups

using a subset of data features). Unless otherwise specified, "model performance" refers to the predictive accuracy for classification tasks, and the mean squared prediction error for regression tasks.

Most prior work on static pruning relies on the intuition that removing the worst trees in the forest will improve the forest's performance. Thus, iteratively removing the worst trees until some stopping criterion is met will theoretically maximize the pruned model performance. For example, Yang et al. (2012) recursively remove the least important tree from the forest, where the least important tree is calculated using the margin criterion. Zhang & Wang (2009) iteratively prune the trees that are the least accurate or the most similar to other trees in the forest. Caruana et al. (2004) greedily add the best trees[1] from the initial forest to the final ensemble until some predefined stopping criterion is met. Giffon et al. (2020) iteratively update the weights for each tree in the forest until the linear combination of the selected trees' predictions approximate the true values.

However, all these proposals have the same issue: they all iteratively compare each tree with all other trees in the forest, then remove or add the one tree that maximizes some target quantity. Ng et al. (1997) cautions against this, as when there are too many hypotheses – or potential trees to include or exclude – selecting the one that maximizes an arbitrary criterion may lead to overfitting, an instance of the "p-hacking" problem. In fact, Tang et al. (2006) find that pruning the ensemble to specifically maximize tree diversity usually does not result in the optimal ensemble, and Caruana et al. (2004) notice that poorly designed or naive static pruning algorithms can lead to overfitting.

As such, the proposed methods will differ from prior research; instead of explicitly constructing an ensemble that maximizes a criterion, tree slices and pruned groups instead leverage the ensemble's inherent ranking structure as a basis for static forest pruning.

## 1.1 Tree Slices

As seen in Figure 1(a), the *tree slicing* approach sorts all the trees using their performance on a subset of training data; a continuous range of trees – or a "slice" – is then selected as the final ensemble. This methodology is motivated by the idea that a sub-ensemble consisting of a continuous range of trees will have greater tree diversity than traditional static pruning approaches. Its performance should be similar to the original ensemble, while accruing major computational advantages during inference.

A "slice" of the ensemble then consists of the trees of rank $r$ through $s$ in the sorted ensemble. While it can be tempting to set $r = n - 1$ and $s = n$, where $n$ is the number of trees in the ensemble, the above

---

[1]Best is calculated using accuracy, cross entropy, mean precision, or ROC area.

concerns regarding overfitting and p-hacking oftentimes justify a different choice for these hyperparameters. It is important to ensure a large diversity of trees in the ensemble.

## 1.2 Pruned Groups

The *pruned groups* approach groups together trees with similar performances; trees that predict similarly are put together. These groups form a representative sample of the trees in the random forest, where each group represents one type of tree in the total forest. An overview of this process is depicted in Figure 1(b).

This methodology is motivated by the idea that trees with similar predictions on the same training data will likely predict similarly to each other on new data; thus, it would be more efficient to use one representative tree from every group instead of using every tree from every group (ie, using the original forest). Because groups sizes are often unequal, the prediction that a representative tree makes is weighted by the number of trees in its group. This methodology reflects our concern with p-hacking, as it uses an intelligent sampling paradigm to reduce the number of trees, instead of simply dropping the worst performing trees.

This process is somewhat similar to the *Statistically Equivalent Signatures (SES)* algorithm, where multiple equivalent feature subsets (or signatures) are constructed that have similar predictive behavior (Tsamardinos et al., 2012; Lagani et al., 2016). For us, the grouping is of trees rather than of features. See Appendix A for an example of how the pruned groups procedure is run.

## 2 Methodology

We propose two methodologies for static random forest pruning: tree slices and pruned groups. As with other static pruning methods, these methodologies are a post-processing step after the initial random forest generation.

## 2.1 Tree Slices

The procedure for tree slices is described in Algorithm 1.

---
**Algorithm 1** Tree Slice

---
**Require:** $r$ and $s$ such that $0 < r < s <$ Number of Trees in Forest
 1: Divide the training dataset into three parts:
 2:     (a) Training dataset for random forest generation
 3:     (b) Training dataset for sorting trees by performance
 4:     (c) Training dataset for validation of the tree slicing process
 5: Create random forest using dataset (a)
 6: Do classification or regression on each tree using dataset (b), then sort the trees based on their performance
 7: Retain all trees between rank $r$ and $s$. These form the new forest, to be used in all future predictions. Validate performance on dataset (c)

---

As aforementioned, "performance" refers to the rate of correct prediction for classification tasks, and the mean squared prediction error for regression tasks.

## 2.2 Pruned Groups

The procedure for pruned groups is described in Algorithm 2, and an example is provided in Appendix A.

The distance used for "K-Means clustering" is the Euclidean distance between any pair of trees, where the position of tree $t$ is the $n$-dimensional coordinate $(x_1, x_2, ..., x_n)$, where $x_i$ is tree $t$'s predicted value for Point $i$ from dataset (b).

---

**Algorithm 2** Pruned Groups

---

**Require:** Group Selector $g$, Tree Selector $t$
 1: Divide the training dataset into three parts:
 2:     (a) Training dataset for random forest generation
 3:     (b) Training dataset for group creation
 4:     (c) Training dataset for group filters
 5: Create random forest using dataset (a)
 6: Do classification or regression on each tree using dataset (b)
 7: Group together similar trees using K-Means clustering with the predictions from (6) as input. Set the number of clusters $k$ to the square root of the number of trees.
 8: Using group selector $g$, determine which groups to select
 9: Using tree selector $t$, determine which trees to select from the chosen groups
10: To predict, take the weighted mode or mean of the selected trees, where weight is equal to the number of trees in the group it was taken from divided by the number of trees taken from that group

---

Table 1: Overview of Utilized Datasets

| | **Covertype** | **Human Activity** | **SV Census** | **Vegas** |
|---|---|---|---|---|
| Description | Forest Coverage in the United States | Smartphone Activity Recognition | Wage Census Data | Vegas Strip Online Reviews |
| Source | Scikit Package[2] | UCI Machine Learning Repository[3] | qeML Package[4] | UCI Machine Learning Repository[5] |
| Task | Classification | Classification | Regression | Regression |
| Number of Features[6] | 54 | 561 | 8 | 121 |
| Number of Classes | 7 | 6 | - | - |
| Number of Instances (Dataset Size) | 250 | 500 | 800 | 504 |
| Size of Dataset (a) | 120 | 240 | 384 | 241 |
| Size of Dataset (b) | 80 | 160 | 256 | 162 |
| Size of Dataset (c) | 30 | 60 | 96 | 60 |
| Size of Validation Dataset | 20 | 40 | 64 | 41 |

Examples of a group selector ($g$) include: groups with at least $l$ trees, groups with an average accuracy above $m$ percent, or groups with an average loss below $n$. Examples of tree selectors ($t$) include: randomly sampling $o$ trees per group, or sampling $p$ percent of trees per group.

## 3  Experimental Datasets

The performance of these algorithms are evaluated against four datasets, of which *Covertype* and *Human Activity* are used for classification, and *SV Census* and *Vegas* are used for regression. An overview of the datasets are provided in Table 1. While other datasets were used, these four form a representative sample of tree slice and pruned group performance.

The *Covertype* dataset uses environmental features to determine dominant tree types in various sections of the Roosevelt National Forest (Blackard, 1998). The *Human Activity* dataset uses smart watch sensor data to predict if user movement. The *SV Census* dataset utilizes a subset of the 2000 Silicon Valley Census data to predict the annual wages for programmers and engineers. The *Vegas* dataset uses hotel and reviewer statistics to predict *TripAdvisor* Las Vegas hotel ratings.

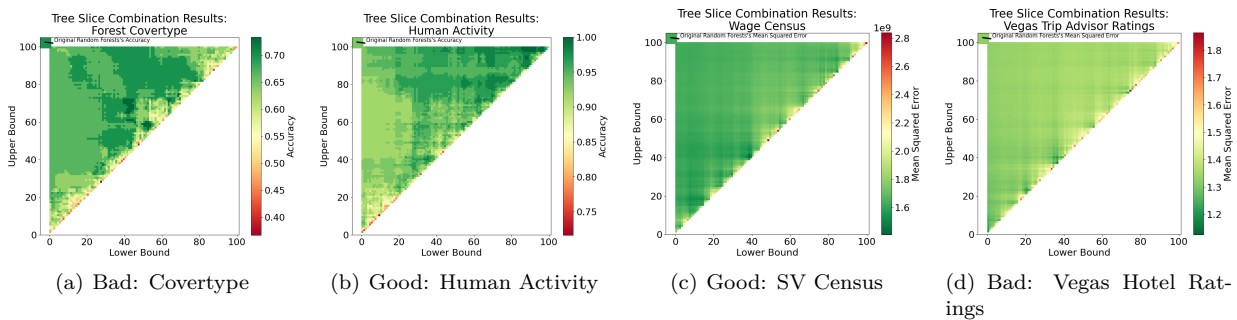

(a) Bad: Covertype    (b) Good: Human Activity    (c) Good: SV Census    (d) Bad: Vegas Hotel Ratings

Figure 2: Tree Slice Results: Performance of Every Slice
To compare, the original random forest's performance is included in the top left square.

# 4    Experimental Performance Results

The innovation for tree slices and pruned groups is that it reduces computation time without affecting model performance. Thus, each methodology is evaluated using two criteria: 1) its performance against the base random forest, and 2) its performance against a new random forest with size equal to the tree slice/pruned group ensemble size.[7] The first criterion ensures that the post-processing step did not significantly alter performance, while the second criterion determines whether this methodology is meaningfully more useful than generating a new, smaller random forest.

## 4.1    Tree Slices

A good tree slice should perform similarly to the full, original forest from which it was derived. With classification, the accuracy should be comparable or better than the random forest, while the mean squared error for regression should be comparable or lower than the random forest.

However, when comparing a tree slice with a new random forest of equal size, the expectation is that tree slices perform *better* than an arbitrarily created random forest with an equal number of trees; the intelligently selected ensemble should be better than the randomly created forest.

To ensure good results and to minimize the effects of noise, all tree slice experiments are repeated 25 times, then their results are averaged together.

### 4.1.1    Criterion 1: Comparison Against Original Random Forest

Tree slices' performance is showcased in Figure 2. The x-axis represents the lower bound $r$ of the slice, while the y-axis represents the upper bound $s$ of the slice – both in percentiles. The color represents the performance of an individual tree slice with the specified upper and lower bounds.

Because tree slices aim to minimize the range of trees while simultaneously maintaining performance and tree diversity, it is theoretically ideal to select slices a bit above the $y = x$ line. As can be seen, Figures 2(b) and (c) represent tree slices that perform well against the original random forest, where several configurations either match or outperform the original random forest. For example with *Human Activity*, a slice from the 65th to 100th percentiles significantly outperformed the performances of the original random forests.

However, tree slices did not perform as well on the *Covertype* and *Vegas* datasets. Figures 2(a) (d) illustrate how some tree slices can actually perform worse than the original random forest. In these, it was generally preferable to take the original random forest over the ensembles created via tree slices, as there was either no performance gain or its performance was somewhat worse.

---

[7]Performance metrics on the same dataset may differ between tree slices and pruned groups because of different averaging techniques and data splits. Thus, tree slice and pruned group metrics from the same dataset should *not* be directly compared between each other.

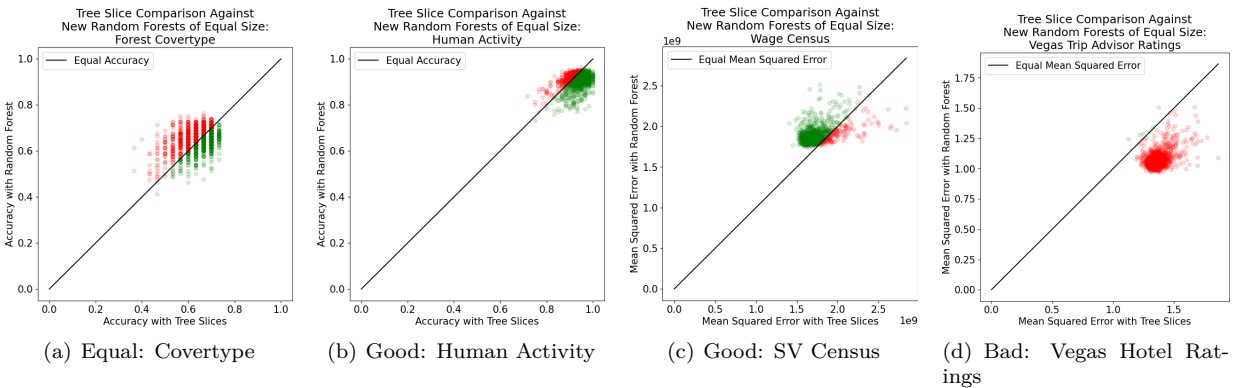

Figure 3: Tree Slice Results: Comparison Against New Random Forests of Equal Size
The $y = x$ reference line is plotted as reference to distinguish between the two halves of the plot.

### 4.1.2 Criterion 2: Comparison Against New Random Forests of Equal Size

Figure 3 compares the performance of a tree slice with the performance of a newly generated random forest of equal size. With classification cases, a good tree slice should be located in the lower right triangle; with regression cases, a good slice should be located in the top left triangle. This is because greater accuracy is desirable with classification, and a lower error rate is desirable with regression.

In general, tree slices outperform the new random forests of equal size. For example, most tree slices from the *Human Activity* and *SV Census* datasets outperform their random forest counterparts. However, this over-performance is not universal. For *Covertype*, tree slicing performs about the same as its random forest counterparts, while *Vegas* tree slices significantly under-perform against their random forest counterparts.

Recall that the underlying principle for tree slices is that *some* slices will outperform the original random forest, and only *one* slice is chosen as the final ensemble. Notably, Figure 3 only showcases the performances of tree slices between the $i$th and 100th percentiles – for all $i$. While these graphs only represent a subset of tree slices, it is sufficient to demonstrate that for many datasets, there exists a type of tree slice that routinely outperforms new forests of equal size. Specifically with the *Human Activity* and *SV Census* datasets, it explicitly demonstrates that tree slices can match the second criterion.

### 4.1.3 Evaluation

The tree slices trained on the *Human Activity* and *SV Census* datasets match the expectations set out in criteria 1 and 2 above. *Covertype* and *Vegas* did not.

The *Covertype* dataset contains a large number of features (54), and only 120 records are provided to create the forest (see Table 1). This suggests that there are not enough records to create a robust tree classifier that can sufficiently leverage all 54 features. As seen along the $y = x$ line of Figure 2(a), most of the trees have very poor individual accuracy. Because all the component trees are poorly constructed in different ways, the *Covertype* forests generally perform better with larger ensemble sizes – instead of via a smarter ensemble. This is also evident with Figure 3(a); the insufficient training data means that this supposedly "smarter" ensemble is just another ensemble of random, poorly-constructed trees. This contrasts with *Human Activity*; with 561 features and more features than records, it still matches the aforementioned expectations. This suggests that unlike with the *Human Activity* dataset, many of *Covertype*'s features are likely important for classification; more training data is needed to create trees that sufficiently leverage the data's features.

For the *Vegas* dataset, when considering Figure 2(d), very few tree slices match the original random forest, and those that did seem to only do so because it includes a few trees that happen to outperform their neighbors. As such, the poor performance for criterion 2 (Figure 3(d)) makes sense; tree slices simply performed poorly. A potential explanation could be that the tree slices are sorted too well, such that similar

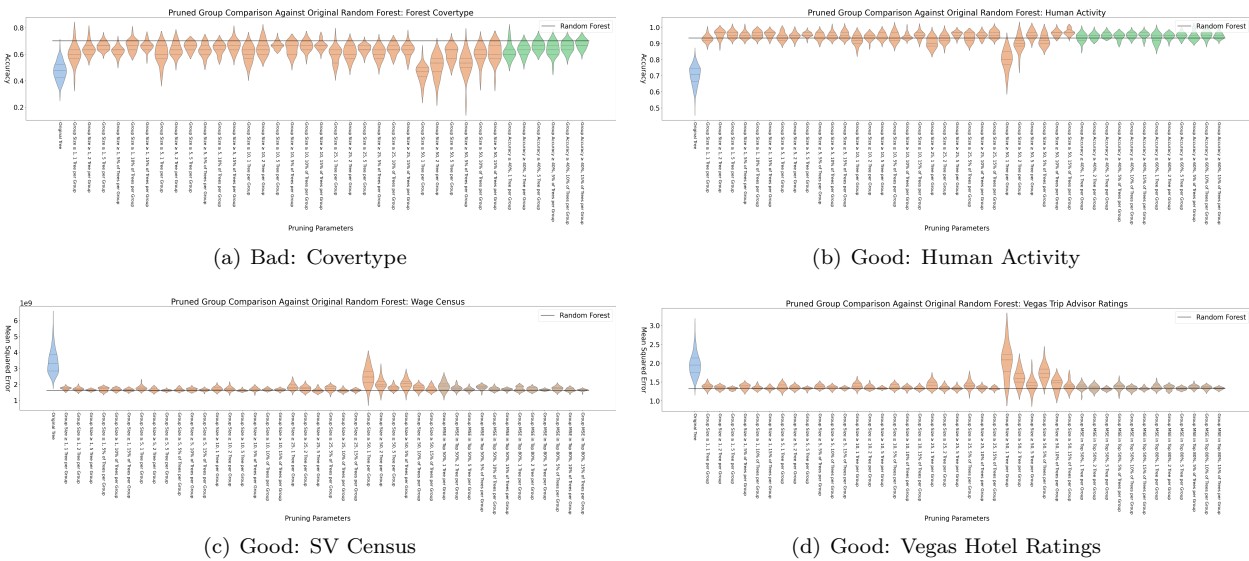

(a) Bad: Covertype

(b) Good: Human Activity

(c) Good: SV Census

(d) Good: Vegas Hotel Ratings

Figure 4: Pruned Groups Results: Performance of Various Criteria

trees are very close together, and those that perform dissimilarly are ordered very far apart. Thus, each tree slice is unlikely to capture a diverse set of trees; in contrast, a newly generated random forest would have more diversity and outperform the diversity-lacking tree slices.

## 4.2 Pruned Groups

Like tree slices, a good pruned group should perform comparably to the random forest it was extracted from, and better than a new random forest with the same number of trees. To ensure consistent results, once the original random forest is generated, 25 pruned groups are created from this one random forest; the results from all 25 groups are plotted in Figure 4, and the 25 averaged results are then plotted in Figure 4.2.2.

### 4.2.1 Criterion 1: Comparison Against Original Random Forest

Figure 4 is a violin plot of the pruned groups' results. The left-most blue violin plot represents the performance of the individual trees that make up the random forest, and the black line is the performance of the random forest generated via said trees. Each subsequent violin plot represents the range of performances based on different filtering criteria (group selector $g$ and tree selector $t$).[8] Because a high accuracy is desirable for classification, violin plots that are equal to or higher than the black line show desirable results; because a low error rate is desired for regression, violin plots that are equivalent or lower than these black lines are desirable. An ideal filtering criteria has small variance and is centered on the black line.

For pruned groups on the *Human Activity*, *SVCensus* and *Vegas* datasets, their performances in Figure 4 generally match or exceed their original random forests' performance. This is expected, as a pruned group is designed as a representative sample of the original random forest. In contrast, *Covertype* consistently performed worse than its original random forest – regardless of grouping and filtering criteria.

A key observation across all datasets is that most pruned groups from the same dataset perform similarly with similar medians, ranges, and distributions – except for the few with a large variance and generally poor performance.[9] These edge cases occur when the filtering condition is too stringent and results in very few groups being selected and/or very few trees being selected from each group. In these situations, only a few trees (sometimes just one tree) are used to represent an entire forest. This is not a representative sample,

---

[8]As aforementioned, there is a range of performance for the subsequent violin plots because each pruned group configuration is repeated 25 times.

[9]E.g.: Human Activity in Figure 4(b), where *Group Size ≥ 50 and 1 tree is selected per group*

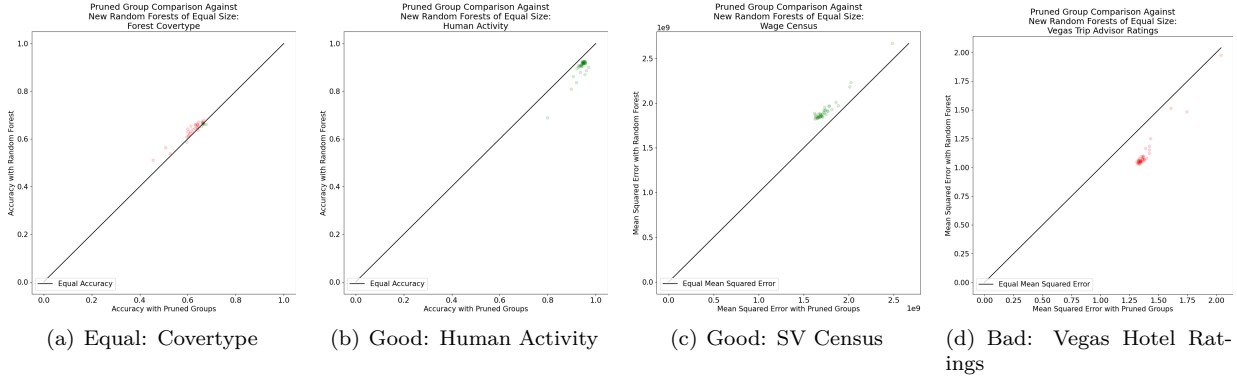

Figure 5: Pruned Groups Results: Comparison Against New Random Forests of Equal Size
The $y = x$ reference line is plotted as reference to distinguish between the two halves of the plot.

so these small ensembles perform worse than the original random forest, and have significant performance variation when the process is repeated.[10] As such, some filtering conditions are not suitable for certain datasets and random forests, as they exclude significant groups and/or utilize an insufficient number of trees. Conditions need to be tailored to the specific experiment and dataset in order to ensure good performance.

If the aforementioned edge cases are excluded from all the plots in Figure 4, all the remaining plots have roughly the same distribution. Recall that there are two main variations for selecting which groups to choose, and two main ways to select how many trees to select per group (see Section 2.2). Considering how different filtering techniques all produce similarly performing pruned groups (with regards to averages, upper/lower bounds, and distribution shape), it suggests that extensively testing different filtering techniques is unnecessary because they all arrive at the same functional result – regardless of the quality of said result. Checking that a set of conditions do not form an edge case should be sufficient in most circumstances.

### 4.2.2 Criterion 2: Comparison Against New Random Forests of Equal Size

The same type of scatter plots (as described in Section 4.1.2) are used to evaluate the performance of pruned groups against a brand new forest of equal size.[11]

Based on Figure 4.2.2, *Human Activity* and *SV Census* perform well when compared against the original random forest. For these datasets, pruned groups consistently outperform random forests of equal size; they have lower errors for regression cases and greater accuracy for classification cases.

There are two datasets that under-performed new random forests of equal size. For *Covertype*, pruned groups perform equivalently (or slightly worse) than random forests of the same size; for *Vegas*, pruned groups perform meaningfully worse than new forests of the same size.

### 4.2.3 Evaluation

From these four datasets, there are three primary ways that a dataset can perform, with each of them revealing certain aspects of the dataset and the random forest configuration.

When pruned groups perform similarly against the original random forest and better against a new forest of equal size, this is an example of the procedure performing exactly as expected. This is exemplified by the *Human Activity* and *SV Census* datasets.

---

[10]This is analogous to how the standard deviation of sample means increases as the sample size decreases (Adhikari et al., 2022). Similarly, sampling few trees from the group can have meaningfully different results if different trees are selected each iteration.

[11]This is a not a direct apples-to-apples comparison. Pruned groups predict via weighed means or weighed modes, where the weight is equal to the size of the group each tree was selected from. Naive random forest only uses means and modes. However, this serves as a decent approximation because it keeps the number of utilized trees the same, and thus the number of trees used in the inference step is kept constant.

When pruned groups perform worse than the original random forest but somewhat similarly to a new forest of equal size, this indicates that the original forest was extremely diverse.[12] Because the number of groups is a predetermined hyperparameter, dissimilar trees may be grouped together simply because there are not enough groups to separate every unique type of tree. Since the pruned group procedure assumes every tree within the same group is similar to each other, sampling only a few trees from a non-homogeneous group means that many tree types may be underrepresented in – or simply absent from – the final forest. Thus, applying the pruned group procedure to forests with high diversity and too-few groups will result in a non-representative forest that under-performs against the larger, original random forest. In addition, because groups are non-homogeneous, this means that two trees selected from one group can share minimal characteristics and are comparable to any two randomly generated trees; as such, the non-representative forest the procedure creates is functionally similar to a new random forest of equal size – and these forests would be indistinguishable from each other. This behavior is exactly reflected in the *Covertype* dataset.

The third scenario is when pruned groups perform well on the original random forest, yet poorly against new random forests of equal size. This indicates that the pruned groups are accurately emulating the performance of the original, large forest – yet the pruned groups under-performing the new, smaller forests implies that a smaller random forest can outperform the original, large forest. This precisely describes the *Vegas* results. This means that the original random forest is not a good ensemble and that a new initial forest with better performance should be re-generated. In fact, when the pruned groups procedure is rerun on the *Vegas* dataset with a different random seed, the pruned groups matched the original random forest's performance *and* outperformed new forests of equal size (see Appendix B). In short, when the initial random forest is good, pruned groups for the *Vegas* dataset performed as well as the *Human Activity* and *SV Census* datasets.

Overall, this indicates that regardless of how precisely the pruned group algorithm is tuned (with regards to its group selector $g$ and tree selector $t$), it is imperative that the original random forest also performs well. For *Covertype*, more data can improve its performance. For *Vegas*, regenerating the forest with different seeds did improve its performance. Thus, if the original forest is sufficiently good, then the pruned group procedure can meaningfully prune the number of trees in the forest without negatively affecting performance.

## 5 Experimental Class Probability Results

For typical classification procedures, models classify an input into one of many classes. Class probability alters the model output by instead outputting the probability that the data is of each class. These values are useful in determining a model's confidence in its outputs.

Tree slices and pruned groups both significantly alter the composition of trees in the forest. Thus, it is important to consider both the accuracy of these pruned ensembles *and* whether these ensembles are similarly confident in their outputs as the original random forest. This section is *not* concerned if these ensembles output accurate class probabilities; it only evaluates if tree slices and pruned groups yield similar class probabilities to their original random forests.

To quantify class probability differences between random forest and tree slice/pruned group performances, two metrics are used: cross-entropy loss[13] and the Kendall's tau coefficient[14]. The ground truth are the metrics calculated from the original random forest's class probability; they are compared against the metrics calculated from the tree slice/pruned group's class probability. If the two ensembles have similar class probabilities, then the absolute difference between their metrics should be small.

To test, this evaluation is carried out on both classification datasets: *Covertype* and *Human Activity*. The exact procedure and algorithm used to carry out these calculations are described in Appendix C.

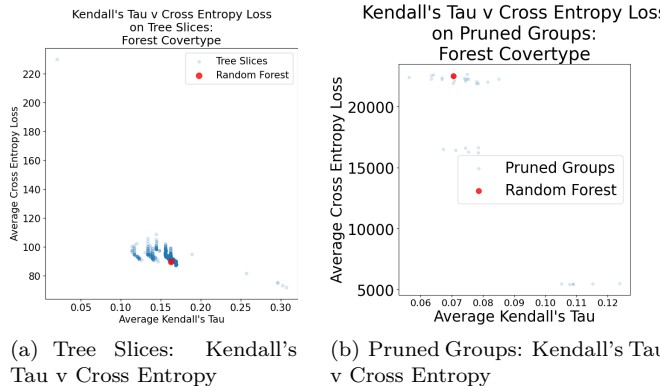

Figure 6: Kendall's Tau and Cross Entropy for Tree Slices and Pruned Groups for Covertype

Table 2: 95% Confidence Intervals for *Covertype*

| Algorithm | Metric | Interval | Target |
|---|---|---|---|
| Tree Slices | Kendall's Tau | [0.143, 0.169] | 0.163 |
| Tree Slices | Cross Entropy | [88.924, 96.154] | 89.851 |
| Pruned Group | Kendall's Tau | [0.062, 0.116] | 0.071 |
| Pruned Group | Cross Entropy | [5441.574, 22636.533] | 22517.236 |

## 5.1 Covertype

The *Covertype* dataset has 7 classes. The cross entropy and Kendall's tau distributions are shown in Figure 6. The red dot represents the cross entropy loss and the Kendall's tau coefficient for the original random forest. Visual inspection of both tree slices and pruned groups show that the metrics for the original random forest are well contained within their distributions. This similarity between the random forest's metrics and the tree slice/pruned group's metrics are quantified with a 95% confidence interval in Table 2. Because all random forest's metrics are all captured within the 95% interval, it suggests that the data could have occurred under the null hypothesis – specifically, that the random forest's metric belongs to the same distribution as the distribution of tree slice/pruned group metrics.

Note that while there are some pruned groups' metrics that are district from the random forest's metrics (see Figure 6(b)). While they are a minority, it indicates that while pruned groups and tree slices generally preserve the class probabilities during their procedures, it is possible for these algorithms to create a new ensemble with noticeably different class probabilities.

## 5.2 Human Activity Index

The *Human Activity* dataset has 6 classes. Figure 7 showcases the cross entropy and Kendall's tau distributions. From visual inspection, the random forest metrics are contained within the distributions. To quantify this, the confidence intervals for the metrics are calculated. As seen in 3, every original random forest metric is contained within its corresponding 95% confidence interval. This means that the original random forest is not statistically distinct enough from the tree slices/pruned group ensembles to conclude they came from different distributions.

---

[12] Diversity can be a consequence of the dataset having too many features and too few instances; the forest will have a large number of diverse, low-quality trees.

[13] Cross-entropy loss measures the difference between two classification distributions; its range is from 0 to infinity.

[14] The Kendall's tau coefficient measures the association between 2 quantities based on concordances and discordances between observations pairs (University, 2024); its range is from -1 to 1.

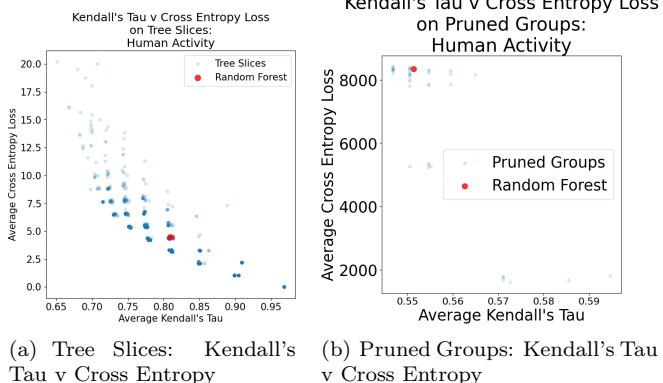

Figure 7: Kendall's Tau and Cross Entropy for Tree Slices and Pruned Groups for Human Activity

Table 3: 95% Confidence Intervals for *Human Activity*

| Algorithm | Metric | Interval | Target |
|---|---|---|---|
| Tree Slices | Kendall's Tau | [0.733, 0.905] | 0.163 |
| Tree Slices | Cross Entropy | [1.060, 7.775] | 4.469 |
| Pruned Group | Kendall's Tau | [0.547, 0.585] | 0.551 |
| Pruned Group | Cross Entropy | [1,669.547, 8,403.960] | 8,339.879 |

For *Human Activity*'s tree slice metrics, there is significant variation; this suggests that there is some instability with these values. While each tree slice may have similar class probability metrics to the original random forest, the metrics between any two tree slices can have significant differences. This is potentially because the tree slice procedure is not designed to specifically guarantee diversity, so different tree slices can be composed of significantly different decision trees; while the predictive accuracy is generally retained (see Section 4.1), ensembles with different types of decision trees will be less likely to produce similar class probability distributions.

On the other hand, the probability distributions are very stable for pruned groups. This is likely because pruned groups are constructed to specifically maintain the predictive diversity of the original random forest via the tree grouping step. As such, the weighted ensemble produced by the pruned group more consistently matches the probability distribution of the random forest.

### 5.3 Evaluation

Based on these datasets, regardless of if the original random forest outputted good class probabilities, tree slices and pruned groups both preserve the baseline random forest's class probabilities. Due to how tree slices are constructed, there is the potential for greater variation in both cross entropy and Kendall's tau coefficients; in contrast, pruned groups have minimal variations in cross entropy and Kendall's tau coefficient.

## 6 P-Hacking Concerns

P-hacking is the practice of repeatedly selecting a subset of variables until a desired outcome is achieved or a performance criterion is met (Bruns & Ioannidis, 2016). Ng et al. (1997) describes this concept like picking a "poor hypothesis that has fit the ... data well 'just by chance'".

Consider 20 people who answer a large survey about their likes and dislikes; from this data, it is found that the number of comedy movies watched and their astrological sign are strongly correlated. As such, it is concluded that these two features are correlated to each other even though this correlation was the result

of pure chance; less significant relationships between every other feature pair are ignored. If this survey is repeated with 20 new people, this specific relationship is unlikely to reappear. This is an example of p-hacking, where relationships are cherry-picked to prove some sort of arbitrary relationship. Similarly, prior static pruning methodologies that seek to only maximize performance raises concerns of p-hacking. There is no guarantee that the subset of trees is truly the best and will generalize well; there is a chance that the random choice of trees performs well only by chance.

For tree slicing, the models generalize well to the validation set. While this does not necessarily resolve all p-hacking concerns, it demonstrates that even though the methodology uses accuracy or error as the sorting criterion, the resulting ensemble predicts in a predictable manner.

For pruned groups, the concerns regarding p-hacking are minimized further because the groups are not created to maximize accuracy or minimize loss; groups are created based on prediction patterns and used to form a representative sample of the entire forest. Accuracy and error are only optionally used for determining which groups to select – but their use can be removed entirely because other non-metric-based techniques exist that produce similar results.

## 7 Discussion and Areas of Future Development

While the two methodologies developed here show promising results in both classification and regression cases for many datasets, they perform less well in other instances. A theoretical analysis of both these techniques can provide further insight into the specific situations that cause performance variations.

There can also be further experiments in the $p >> n$ case, where the number of features is significantly greater than the number of instances. The *Human Activity* dataset tests this, but a more extreme example like a human genomics dataset, where there are thousands of genes (features) for only a few subjects (instances), can be an even more rigorous stress test to ensure that tree slices and pruned groups will still function as expected in these extreme circumstances.

For tree slices, there should be research into striding: selecting only every $m$-th tree in between the specified lower and upper bounds. This can be a computationally simpler alternative to pruned groups, as it is an intelligent sampling technique that does not directly rely on accuracy or loss for determining which trees to select.

For pruned groups, there should be experimentation into different metrics for quantifying tree similarity. The pruned groups algorithm in Algorithm 2 uses the Euclidean distance between two trees, but Kulkarni & Sinha (2012) describe a different metric that uses the "correlation between the predicted outcomes of two trees," while Perner (2013) proposes a "rule set" technique that quantifies the similarity of all nodes between two trees. If these metrics can more effectively group similar trees, then it could improve the pruned groups algorithm.

Another improvement to the pruned groups algorithm is to dynamically determine the optimal number of groups. The number of groups is currently hardcoded, which means pruned groups are less effective when grouping forests with large tree diversity (see Section 4.2.3's discussion on the *Covertype* datasets).

## 8 Conclusion

Tree slices and pruned groups are empirically shown to be efficient ways of reducing random forest ensemble sizes without significantly affecting performance. By matching the performance of the original random forest and outperforming naive random forests of equal size, these methodologies can perform exceptionally well in both classification and regression cases. These methodologies also largely preserve class probabilities for classification cases. However, the fact that these procedures can – at times – create noticeably under-performing ensembles means that future work still needs to be done to make sense of this performance difference.

Regardless, as it currently stands, these methodologies can be an effective post-processing step for random forests that optimize performance for future classification and regression tasks.

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

# A    Example of the *Pruned Groups* Procedure

Consider a random forest with 6 trees. These 6 trees have made these predictions in Table 4; these predictions are also visualized in Figure 8.

Table 4: Example Random Forest Predictions

| Tree | Point A Predictions | Point B Predictions | Group Number |
|---|---|---|---|
| Tree 1 | 2 | 4 | Group 1 |
| Tree 2 | 1.9 | 4.1 | Group 1 |
| Tree 3 | 6.1 | 1 | Group 2 |
| Tree 4 | 6.0 | 1.1 | Group 2 |
| Tree 5 | 5.9 | 0.9 | Group 2 |
| Tree 6 | 6.0 | 1 | Group 2 |

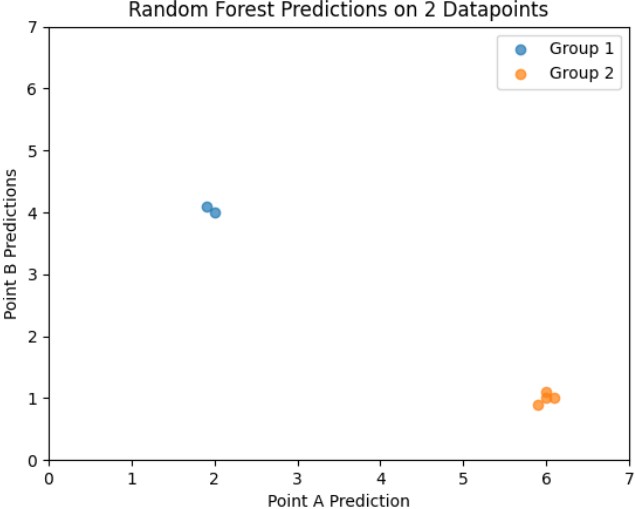

Figure 8: Example Random Forest Predictions

Based on every tree's predictions on Points A and B, Trees 1 and 2 have similar predictions, and Trees 3, 4, 5, and 6 predict similarly to each other. An ideal K-Means clustering of these 6 trees would create these 2 clusters: Group 1 with Trees 1 and 2, and Group 2 with Trees 3, 4, 5, and 6; any tree sampled from either group should be a good approximation of all the other trees in its group. Thus, if one tree is sampled from each group and their predictions are weighted together,[15] the predictions would be a reasonable approximation of the original random forest's predictions.

# B    Rerunning Pruned Groups on the *Vegas* Dataset with Different Seeds

Appendix B highlights how a poor initial random forest can undermine the pruned groups algorithm.

All prior procedures – including the pruned groups procedure on the *Vegas* dataset – use an arbitrarily chosen initial seed of 0. As Figure **??** indicates, the original pruned groups for the *Vegas* dataset perform comparably to the original random forest. As Figure 5(d) indicates, these pruned groups perform worse than

---

[15]The predictions from Group 1's tree will have a weight of 2 because Group 1 has 2 trees; the predictions from Group 2's tree will have a weight of 4 because Group 2 has 4 trees.

new forests of equal size. As aforementioned in Section 4.2.3, this is unexpected, because it means a larger random forest has worse performance than a smaller random forest. This suggests that the original random forest was – by chance – a poor performer.

When the pruned groups procedure is repeated with a different seed (namely, a seed of 1), the output is substantially better. Figure 9 demonstrates that the pruned group algorithm is still able to accurately match the performance of this newly generated large random forest. Importantly however, Figure 10 shows that these pruned groups *also* outperform trees of equal size. A better initial forest makes the resulting pruned groups more effective. In fact, this new performance matches that of the *Human Activity* and *SV Census* datasets as discussed in section 4.2.3, and indicates that the *Vegas* dataset – when properly preprocessed – can match our expectations.

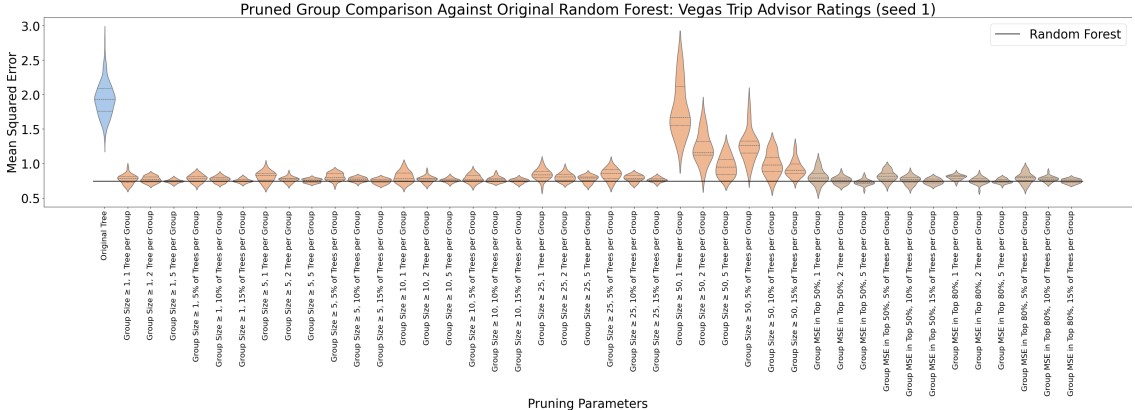

Figure 9: New Pruned Group Results on the *Vegas* Dataset: Performance of Various Criteria

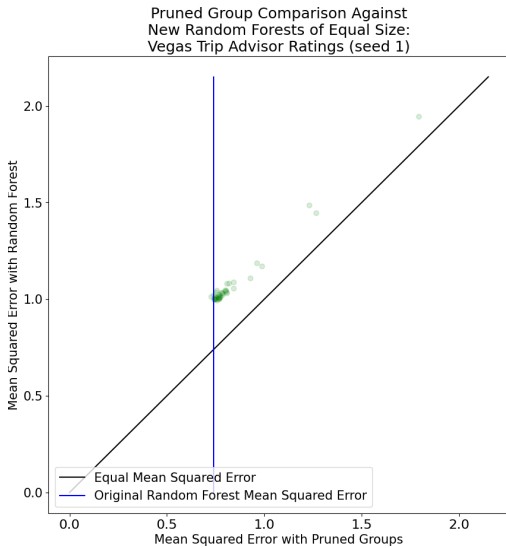

Figure 10: New Pruned Group Results on the *Vegas* Dataset: Comparison Against New Random Forests of Equal Size

As such, whenever a pruned group matches the original random forest but under-performs against equally-sized forests, it suggests that the original forest is a poor model – not that the algorithm is flawed. To improve pruned group performance, the solution is to regenerate the original forest.

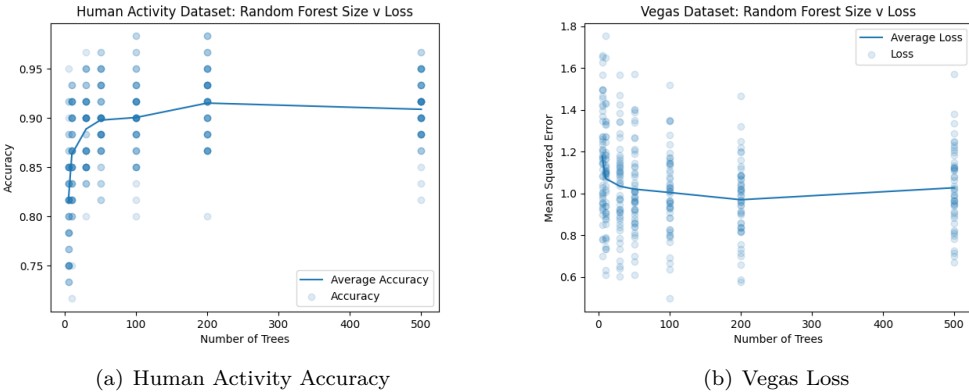

(a) Human Activity Accuracy  (b) Vegas Loss

Figure 11: Comparison Between Random Forest Size and Performance

Recall that the pruned group procedure seeks to replicate a forest's performance with fewer trees; it is not designed for performance improvements. Thus, the original forest must be good. The primary driving issue with the *Vegas* dataset is that forests generated with this dataset do not meaningfully improve with more trees. Figure 11(b) shows the performance of 350 total forests with varying numbers of trees; more trees did not meaningfully improve performance. Because a high-performing model is desired, a model that outperforms forests with fewer trees ought to be selected; with the *Vegas* dataset, this means this initial forest must *also* outperform most forests of the same size. When this advice is heeded, good pruned groups like in Figures 12(a), 12(b), and 12(c) can occur. When any arbitrary forest is selected, bad pruned groups like in Figures 12(d), 12(e), and 12(f) can occur.

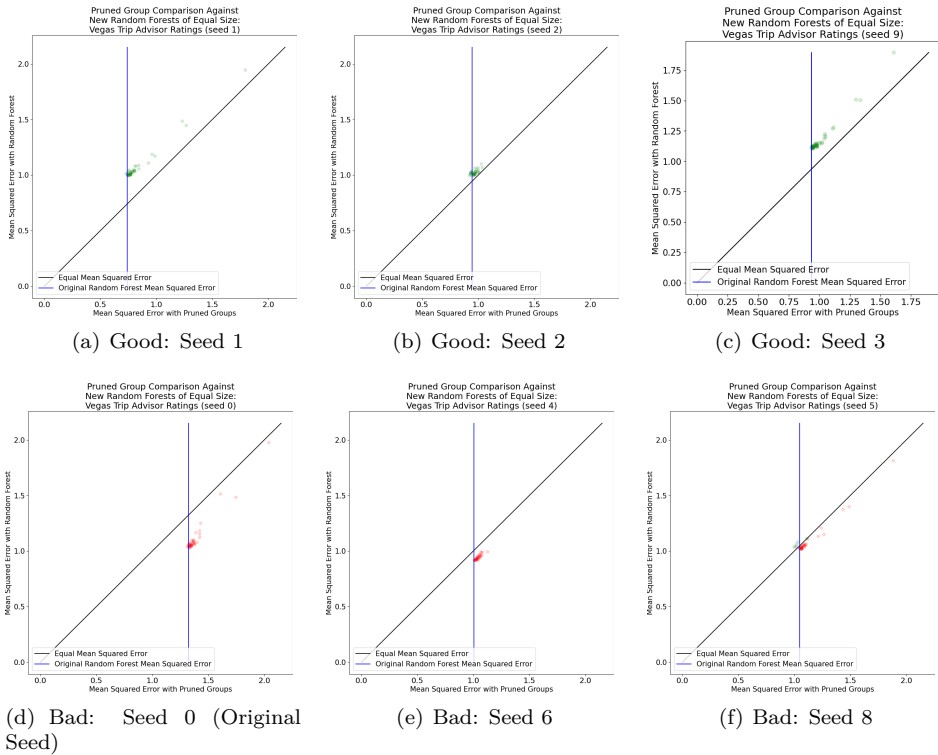

(a) Good: Seed 1  (b) Good: Seed 2  (c) Good: Seed 3

(d) Bad: Seed 0 (Original Seed)  (e) Bad: Seed 6  (f) Bad: Seed 8

Figure 12: Pruned Group with Different Seeds Results: Comparison Against New Random Forests of Equal Size

Of note, this preprocessing stage for selecting a good forest often requires few (if any) forest re-generations. For example, the *Human Activity* dataset's random forest performance in Figure 11(a) demonstrates that as the size of the forest increases, accuracy meaningfully increases until it plateaus after 50 trees. Because the pruned groups generated by the *Human Activity* dataset have a median of 35 trees, and over 75% of pruned groups have fewer than 50 trees, it is expected that most pruned groups generated from a random forest of 500 trees will outperform smaller forests of size equal to the pruned groups. In fact, the pruned groups showcased in Figures 4(b) and 5(b) required no random forest re-generations.

In short, the original random forest must be first validated to have good performance before any static pruning procedure should be carried out. Failure to do so can result in poor pruned group results.

## C   Algorithms for Calculating Class Probability Metrics

Appendix C describes how the raw data for class probabilities are collected, processed, and used to output metrics and graphs.

Each dataset has a list of forests that have been generated. This list corresponds to either a set of tree slices or pruned groups. Algorithm 3 calculates the cross entropy and Kendall's tau coefficient for all provided forests, then outputs the graphs for this data. It uses Algorithm 4 to calculate the class probability for one forest, Algorithm 5 to calculate the cross entropy metric, and Algorithm 6 to calculate the Kendall's tau coefficient.

Algorithm 4 takes as input any forest and any dataset, then outputs the matrix *class_prob_matrix* that contains the class probabilities. The keys of *class_prob_matrix* are the true classes, and the keys for *class_prob_matrix[c]* are the predicted classes for class *c*. The values ($class\_prob\_matrix[c][\hat{c}]$) represent the total weight of each true and predicted class combination.

The formula for cross entropy is: $cross\_entropy = -\sum(p_k * \log(p_k)) + \sum(p_k * \log(\frac{p_k}{q_k}))$. $p_k$ is the ground truth distribution and $q_k$ is the predicted class distribution. This cross entropy implementation uses *SciPy*'s *entropy* implementation.[16]

The formula for Kendall's tau is: $tau = \frac{P-Q}{\sqrt{(P+Q+T)*(P+Q+U)}}$. $P$ is the number of concordant pairs, $Q$ the number of discordant pairs, $T$ the number of ties only in the first array, and $U$ the number of ties only in the second array. If a tie occurs for the same pair in both arrays, it is not added to either $T$ or $U$. The Kendall's tau implementation uses *SciPy*'s default $tau_b$ implementation.[17]

---

**Algorithm 3** Class Probability for List of Forests

**Require:** $r$ and $p$, where $r$ is a random forest and $p$ is the list of forests created from $r$
**Require:** dataset $D$
 1: $cross\_entropy\_records \leftarrow []$
 2: $tau\_records \leftarrow []$
 3: $ensembles \leftarrow p + [r]$ (ie, append $r$ to $p$)
 4: **for** forest $f$ in $ensembles$:
 5:     $curr\_class\_prob\_matrix \leftarrow$ Algorithm 4 on $F = f$ and $D = D$
 6:     $cross\_entropy\_records$.append(Algorithm 5 on $curr\_class\_prob\_matrix$)
 7:     $tau\_records$.append(Algorithm 6 on $curr\_class\_prob\_matrix$)
 8: **return** scatter($cross\_entropy\_records$, $tau\_records$)
 9: **return** histogram($cross\_entropy\_records$)
10: **return** histogram($tau\_records$)

---

[16]https://docs.scipy.org/doc/scipy/reference/generated/scipy.stats.entropy.html
[17]https://docs.scipy.org/doc/scipy/reference/generated/scipy.stats.kendalltau.html

---

**Algorithm 4** Calculating Class Probabilities for One Forest

---

**Require:** Forest $F$ and Dataset $D$
 1: Initialize empty dictionary $class\_prob\_matrix$, with values 0
 2: For data $d$ in $D$:
 3:     Let $f$ be the features of $d$ and $c$ be the class of $d$
 4:     For each tree $t$ in $F$:
 5:         Let $w$ be the weight associated with $t$. If no weight is specified, let $w = 1$
 6:         Classify feature $f$ using tree $t$. Let the class be $\hat{c}$
 7:         $class\_prob\_matrix[c][\hat{c}] += w$
 8: **return** $class\_prob\_matrix$

---

**Algorithm 5** Cross Entropy Helper

---

**Require:** $class\_prob\_matrix$ from Algorithm 4
 1: $n \leftarrow$ number of classes in $class\_prob\_matrix$
 2: $loss \leftarrow 0$
 3: for $i$ in $[0, ..., n-1]$:
 4:     $y_i\_true \leftarrow [0] * n$
 5:     $y_i\_true[i] \leftarrow 1$
 6:     $y_i\_weight = sum(class\_prob\_matrix[i])$
 7:     if $y_i\_weight = 0$:
 8:         continue
 9:     $y_i\_class_probs \leftarrow \frac{class\_prob\_matrix[i]}{y_i\_weight}$
10:     $curr\_loss \leftarrow entropy(y_i\_true) + entropy(y_i\_true, y_i\_class_probs)$  ▷ This $entropy$ call calls scipy's entropy function
11:     $loss += (curr\_loss * y_i\_weight)$
12: **return** $loss$

---

**Algorithm 6** Kendall's Tau Helper

---

**Require:** $class\_prob\_matrix$ from Algorithm 4
 1: $n \leftarrow$ number of classes in $class\_prob\_matrix$
 2: $ground\_truth\_matrix \leftarrow zeros(n, n)$
 3: for $c$ in $class\_prob\_matrix$:
 4:     $ground\_truth\_matrix[c][c] \leftarrow 1$
 5: **return** $kendalltau(ground\_truth\_matrix, class\_prob\_matrix)$     ▷ This $kendalltau$ call calls scipy's kendalltau function

---

