# OpenReview forum: "Good Trees: Pruning Random Forests Without Compromise"
_TMLR — Rejected by TMLR_

### Review · Reviewer_fJeW · 2025-01-28

**Summary Of Contributions:**

This paper describes two strategies to prune ensembles of trees generated by random forest. Although the methods are more general and they can be applied to other ensembles. The first strategy simply sorts the trees according to their prediction accuracy and selects a subset from the best tree to the works tree. The second strategy performs clustering on the trees predictions and selects a representative subset of trees from each cluster according to several criteria. These methods are evaluated on 4 datasets and compared to the original random forest ensemble and to random forest ensembles of the same size as the pruned ensemble.

**Audience:**

Yes

**Claims And Evidence:**

No

**Requested Changes:**

The paper is missing important related literature to static and dynamic pruning in ensembles of classifiers. In particular, the references:

Statistical instance-based pruning in ensembles of independent classifiers
D Hernández-Lobato, G Martinez-Munoz, A Suárez
IEEE Transactions on Pattern Analysis and Machine Intelligence 31 (2), 364-369

An analysis of ensemble pruning techniques based on ordered aggregation
G Martinez-Munoz, D Hernández-Lobato, A Suárez
IEEE Transactions on Pattern Analysis and Machine Intelligence 31 (2), 245-259

Some figure references are wrong. E.g. those mentioning fig. 5.

Cross-entropy loss13 and the Kendall’s tau coefficient are not properly explained. They should be explained.

The experimental section has to be improved a lot by:

1 - Including more datasets in the experiments. At least try to consider a similar setup as the one described in the references above.

2 - Include other relevant ensemble pruning strategies from the literature. These may be the ones mentioned in the references above, but also others from the literature such as the ones mentioned by the authors in their paper.

3 - Provide a method to determine the hyper-parameters of the proposed techniques and give results for those hyper-parameter values instead of giving results for all hyper-parameters which favors methods with a large number of hyper-parameters.

4 - Ensure that the comparison is fair by using all available data to build the full random forest ensemble and the random forest the authors compare results with. Currently it is not clear if this is the case. If the full ensemble of random forest is not using the validation data for training the comparison is unfair.

**Strengths And Weaknesses:**

The strengths of the paper are the simplicity of the proposed methods and the good writing of the manuscript. However, the paper has important drawbacks.

The main point of criticism of this paper is the experimental section. In particular, only 4 datasets seem to be considered which is a small number to draw conclusions. Furthermore, it is not clear or not if the random forest the authors compare results with has used all the available data for training. This is relevant because the proposed methods need a validation set which means that the random forest ensemble is using less data for training and hence will perform worse than the full random forest.

Another point of criticism is that no comparisons with other ensemble pruning methods from the literature are carried out. See the references below and other methods the authors mention in their paper.

Finally, the methods proposed by the authors have several hyper-parameters that must be chosen (for each pruning strategy). It is not clear how to choose them. The authors simply report results for each potential value of these hyper-parameters, which favors always methods with the largest number of hyper-parameters. This questions the significance of the reported results.

---

### Review · Reviewer_67Dd · 2025-02-08

**Summary Of Contributions:**

This paper proposes two approaches, tree slices and pruned groups, designed to reduce the size of a random forest while preserving its original accuracy. There are many previous works on distilling the knowledge of a random forest into a smaller model. I believe this work is in line with these works.

**Audience:**

Yes

**Broader Impact Concerns:**

I believe there are not concerns.

**Claims And Evidence:**

Yes

**Requested Changes:**

1. Include more related works. Distilling the knowledge of a random forest into a smaller model, e.g., a single tree, a neural network, or a smaller forest, or pruning the structure of a forest, have been studied widely. Compare the proposed ideas and their performance with these approaches.

Here are a few references you may find interesting. There are much more in the literature.
[1] Ensemble Selection from Libraries of Models. ICML 2004
[2] Interpreting Tree Ensembles with inTrees. arXiv 2014
[3] EDiT: Interpreting Ensemble Models via Compact Soft Decision Trees. ICDM 2021
[4] Random Forest Pruning Techniques: A Recent Review. Operations Research Forum (2023)

2. Include more datasets. I believe feature-based (or tabular) datasets are generally too small and not suitable as a benchmark if we use only a few datasets. There are many papers introducing a more comprehensive benchmark. Refer to https://arxiv.org/abs/2406.19380

**Strengths And Weaknesses:**

Strengths
1. This paper is written well and easy to follow. The algorithms well describe how the proposed approaches work.
2. The authors present many good insights on the experimental results, analyzing them from various perspectives.

Weaknesses
1. Experiments are weak. There should be more datasets to for meaningful evaluation of the proposed approaches and more competitors which were proposed for a similar purpose.
2. Proposed ideas lack technical novelty. There are no theoretical analysis and technical contributions that can result in promising future works.

---

### Review · Reviewer_3Pot · 2025-02-10

**Summary Of Contributions:**

The authors propose two methods for reducing the number of trees in a random forest ensemble based on tree similarity rather than an optimization criterion, which could risk overfitting. These approaches aim to maintain the predictive performance of the original forest while using fewer trees, thereby improving computational efficiency.

**Audience:**

Yes

**Claims And Evidence:**

No

**Requested Changes:**

- compare against the baseline of selecting from the forest the same number of trees uniform at random
- use a critical difference diagram to report results
- quantify the speed up : comparing the number of selected trees against the accuracy of the resulting forest
- offer practical guidance for tuning: what procedure to use for hyper parameter selection: start/end (and stranding ?) for  Tree Slices and cluster numbers (or distance threshold for DBSCAN type approaches).

**Strengths And Weaknesses:**

The idea is both simple and efficient, making it worthy of exploration. The concept of avoiding direct optimization of a task-related metric or excessive pairwise tree comparisons is particularly intriguing.

However, the key issue lies in how convincingly the authors can demonstrate the effectiveness of their approach. Specifically:
- A crucial baseline for comparison is whether selecting the same number of trees uniformly at random results in significantly worse forests. The authors do not explicitly address this.
- In all cases, it is essential to robustly assess whether the results are statistically significant. To achieve this, the authors should use a critical difference diagram (code available at [scikit-posthocs](https://scikit-posthocs.readthedocs.io/en/latest/generated/scikit_posthocs.critical_difference_diagram.html)) to provide clear evidence that the observed improvements are meaningful.
- Another unexplored aspect is the quantification of the speedup: how much faster is the resulting forest? A graph comparing the number of selected trees against the accuracy of the resulting forest would be valuable, but this relationship is not presented.
- Additionally, a systematic study of the approach's dependence on its hyperparameters is missing. For instance, in the slice approach, how should the start and end of the interval be chosen? In the pruned groups approach, how should the number of clusters be determined? More importantly, how do these choices impact the final accuracy? While the authors provide an exhaustive plot for the slice approach, they do not offer practical guidance on selecting these values. If every possible value must be tested empirically, the increased computational cost of an extensive fitting phase should be weighed against the speedup gained during inference.

---

### Author Response · Authors · 2025-02-24
**Response to Reviewers fJeW, 67Dd, and 3Pot**

Thank you all for your comments and feedback, I'll respond to the major ones below.

(**fJeW**) With regards to related literature, we will add and reference most of the suggested literature.

(**fJeW, 67Dd**) With regards to needing more datasets, we agree. We were unsure how to format the results from multiple datasets, so we opted instead for an extensive analysis of a sample of datasets. Moving forward, we will be taking Reviewer **fJeW**'s advice by adding more datasets and structuring them like the references you provided.

(**67Dd**) With regards to comparing with other ensemble pruning methods, we can try and compare one or two other methods, but its feasibility will still need to be determined because the code may not be available in all cases.

(**fJeW, 3Pot**) With regards to hyperparameter choice, we already briefly discuss this for pruned groups; we say we say in the last paragraph of 4.2.1 and performance is largely independent of hyperparameter choice (group selection g and tree selector t) -- but this can be made more explicit. However more generally, the primary purpose of pruning is to speedup the inference step; so while general guidelines can be developed for tree slices (such as using every tree between the 40th and 60th percentiles), it is still relatively effective to test all potential slice boundaries as a preliminary step before setting on an ideal inference strategy. This means that while hyperparameter choices can be added for tree slices and elaborated for pruned groups, we don't believe an explicit "best" hyperparameter choice is necessary.

(**fJeW**) With regards to using all available data -- yes, we are using all available data; it is split into datasets (a), (b), and (c) for the 3 sub-datasets needed during training/validation. We can make this more explicit.

(**3Pot**) With regards to comparing with a baseline forest, we already do something similar to that in section 4.1.2. and 4.2.2; in these, we compare tree slices/pruned groups with a brand new forest of equal size. Do you think it is necessary to further compare tree slices/pruned groups with a naive, random sample of equal size from the original large forest?

(**3Pot**) With regards to the critical difference diagram and statistical significance testing, we think it would be better to form simultaneous confidence intervals, say via the Scheffe's method.

(**3Pot**) With regards to the speedup, we can add that in.

---

> ### Comment · Reviewer_fJeW · 2025-02-24
> **Random Forest and training Data**
>
> In your response, it is unclear if random forest uses all the available data for training. This includes the training set and also the validation set. Since Random Forest does not need a validation set it can use the validation set as training data. If you are not comparing with random forest trained on the training data plus the validation set, the comparison is unfair.
>
> If a method lacks public code, you can always write your own implementation.
>
> Best.

---

> > ### Author Response · Authors · 2025-03-17
> > **Re: Random Forest and Training Data**
> >
> > It sounds like you are using the term 'validation set' to mean what others call the 'test set' or 'holdout set', opposed to dividing the original dataset into training, test and validation sets. But actually under the second definition, yes, a validation set IS needed to get unbiased results, which are key in our experiments. The reason is that when one looks at the output of the test set, one takes the best hyperparameter combination, which produces a bias. If say random variables X_1 and X_2 both have mean mu, one can show that the mean of min(X_1,X_2) is less than mu.
> >
> > With regards to finding models, we will reach out to the authors for their code. If they can't provide it, we won't be able to replicate their code because published algorithms only provide outlines, not details and implementation.

---

> > > ### Comment · Reviewer_fJeW · 2025-03-17
> > >
> > > Again, in your response, it is unclear if Random Forest uses all the available data for training. This includes the training set and the validation set. Since Random Forest does not need a validation set, it can use the validation set as training data. Could you please clarify this?

---

> > > > ### Author Response · Authors · 2025-04-08
> > > >
> > > > The random forest does not use all available data for training, because a reserves a portion for validation. There is literature that states a validation set is necessary.

---

> > > > > ### Comment · Reviewer_fJeW · 2025-04-08
> > > > >
> > > > > Can you provide references indicating the need for a validation set? What hyper-parameters do you need to tune?

---

### Author Response · Authors · 2025-04-08
**validation sets**

Someone asked for a reference on validation sets. I believe my coauthor supplied one in a Comment here.

But regarding one of the early review comments that using cross-validation unfairly advantaged our methods, I realized last night that there has been some miscommunication this (our fault). I see now that the reviewer was thinking in terms of our methods' ability to reduce runtime at the inference stage, while we had thought he/she meant in terms of accuracy. In the latter setting, our methods are DISadvantaged.

---

> ### Comment · Reviewer_fJeW · 2025-04-08
>
> Can you please introduce here again the reference where it is indicated that random forest requires a validation set to tune hyper-parameters?
>
> My concern is that it seems that your method needs extra data for ensemble pruning besides the data required to create a random forest. Those extra data can be simply used to improve the performance of random forests by adding the extra data to the training set used to create the random forest. I believe that you are not considering this.

---

### Decision · Action_Editor_SJA9 · 2025-03-19

**Recommendation:** Reject

**Comment:**

I provide below the main issues justifying my decision on this submission.

- **Missing Baseline Comparison:** The paper does not compare its methods to even a simple random selection of trees, making it unclear if the proposed approach is truly better.  Comparisons against other SOTA approaches would be also needed.
- **Lack of Statistical Significance Testing:** Reviewers requested critical difference diagrams or similar tests, but the authors did not provide them.
- **Unclear Speedup and Hyperparameter Tuning:** There is no clear evidence of computational speedup, and guidelines for choosing hyperparameters are vague.
- **Fairness of Experiments:** One reviewer pointed out that the comparison may be biased because the proposed method could be using more training data than the baseline.

I would be happy to reconsider this paper for publication is authors explicitly address the above issues in a new resubmission.

**Audience:**

Reviewers think TMLR’s audience would be interested. However, Reviewer 3Pot specifically marks “Audience: No” because the paper seems neither novel enough nor rigorously demonstrated to hold the TMLR reader’s attention.

**Claims And Evidence:**

The reviewers do not find that the paper’s claims are adequately supported by accurate, convincing and clear evidence.  In particular:

• Baseline Comparisons: The reviewers repeatedly ask the authors to compare their pruning methods to a simple random subsample of the original forest but note that the authors have neither performed nor firmly committed to including that comparison. This makes it difficult to tell whether the proposed methods truly surpass a naive baseline.

• Speedup & Hyperparameter Clarity: The paper’s claims include efficiency gains, but reviewers see no detailed or systematic account of the actual speedups. Similarly, the reviewers consistently report that guidelines for selecting hyperparameters (number of clusters in “pruned groups,” start/end boundaries in “tree slices,” etc.) are insufficiently explained for others to replicate or evaluate results fairly.

• Fair Use of Data: One reviewer specifically argues that the authors’ experiments appear to allocate more data to training their proposed approach than to training the baseline random forests, thereby biasing results. Because the submission never clarifies this point, it leaves open the possibility of an “apples-to-oranges” comparison.

In summary, given these omissions—particularly the lack of a strong random baseline comparison,  incomplete evidence about speedups, and unclear or potentially biased experimental setups—the reviewers do not find that the paper’s claims are adequately supported by “accurate, convincing and clear evidence,” as required by TMLR’s policies on acceptance.

**Resubmission Of Major Revision:**

The authors may consider submitting a major revision at a later time.